EMBO
Molecular Medicine

# Compounds producing an effective combinatorial regimen for disruption of HIV-1 latency

Pargol Hashemi[1], Kris Barreto[1], Wendy Bernhard[1], Adam Lomness[1], Nicolette Honson[2], Tom A Pfeifer[2], P Richard Harrigan[3] & Ivan Sadowski[1,*]

## Abstract

**Highly active antiretroviral therapy (HAART) has improved the outlook for the HIV epidemic, but does not provide a cure. The proposed "shock-and-kill" strategy is directed at inducing latent HIV reservoirs, which may then be purged via boosted immune response or targeting infected cells. We describe five novel compounds that are capable of reversing HIV latency without affecting the general T-cell activation state. The new compounds exhibit synergy for reactivation of latent provirus with other latency-reversing agents (LRAs), in particular ingenol-3-angelate/ PEP005. One compound, designated PH02, was efficient at reactivating viral transcription in several cell lines bearing reporter HIV-1 at different integration sites. Furthermore, it was capable of reversing latency in resting CD4[+] T lymphocytes from latently infected aviremic patient cells on HAART, while producing minimal cellular toxicity. The combination of PH02 and PEP005 produces a strong synergistic effect for reactivation, as demonstrated through a quantitative viral outgrowth assay (qVOA), on CD4[+] lymphocytes from HIV-1-infected individuals. We propose that the PH02/ PEP005 combination may represent an effective novel treatment for abrogating persistent HIV-1 infection.**

**Keywords** HIV-1; latency-reversing agent; latent reservoir; shock and kill; sterilizing cure

**Subject Categories** Microbiology, Virology & Host Pathogen Interaction; Pharmacology & Drug Discovery

## Introduction

Despite nearly 36 years of intensive research investigating acquired immune deficiency syndrome (AIDS) and human immunodeficiency virus (HIV), mankind still faces major challenges in developing a cure for this disease. A significant challenge is posed by the ability of the virus to form latent infections in the long-lived memory T-cell population. During replication in CD4[+] T lymphocytes, a major target for HIV-1 infection, a small fraction of activated cells revert to a resting state, $G_0$, in which transcription of chromosomally integrated provirus is repressed, causing the virus to become transcriptionally silenced or latent (Finzi et al, 1997). The highly stable and long-lived resting CD4[+] T cells carrying integrated latent provirus are thought to represent a major barrier to an HIV-1 cure, because the immune system cannot recognize and eliminate these cells from patients (Palmer et al, 2011). Generally, the cessation of therapy results in rapid reappearance of the virus from reactivated latent reservoirs, which renders patients dependent on a lifetime of antiretroviral therapy (Shan & Siliciano, 2013). Therefore, without strategies for eliminating latently infected cells, HAART eventually becomes ineffective, because resistant viral strains emerge and side effects gradually accumulate through the prolonged use of antiretroviral drugs (Gunthard et al, 1998; Demeter et al, 2004; Lyons et al, 2005).

A combination of cellular events contributes to establishment of latent provirus. Transcription from the integrated viral genome depends on the host cell transcriptional machinery. Specifically, signaling pathways stimulated by engagement of the T-cell receptor (TCR) regulate a collection of transcriptional activators bound to cis-elements in the long terminal repeat (LTR) promoter region, which recruit chromatin-remodeling complexes and general transcription factors for RNA polymerase II to initiate viral transcription (Bassuk et al, 1997). In resting memory CD4[+] T lymphocytes, these transactivators are replaced on the LTR by factors that recruit histone deacetylases (HDACs) and histone methyltransferases (HMTs), which cause chromatin condensation and preclude initiation by RNA polymerase II (Sadowski et al, 2008). Some evidence also suggests the presence of microRNAs (miRNAs) that target the 3′ end of HIV-1 mRNA, which causes silencing of the integrated provirus (Huang et al, 2007). An important event in establishment of latency is the loss of the transactivator Tat, a viral protein that regulates transcriptional elongation from the viral promoter (Razooky & Weinberger, 2011; Donahue et al, 2012). This state of non-productive infection, represented by a transcriptionally silenced provirus, is likely due to a combination of these aforementioned mechanisms that function to establish and maintain a persistent latent infection.

1   Biochemistry and Molecular Biology, Molecular Epigenetics, Life Sciences Institute, University of British Columbia, Vancouver, BC, Canada
2   The Centre for Drug Research and Development, Vancouver, BC, Canada
3   BC Centre for Excellence in HIV/AIDS, St. Paul's Hospital, Vancouver, BC, Canada
    *Corresponding author. Tel: +1 6048224524; Fax: +1 6048225227; E-mail: ijs.ubc@gmail.com

A sterilizing HIV cure represented by complete clearance of viral particles and nucleic acids from the patient's body cannot happen unless the latently infected cells are eradicated. One potential HIV cure strategy, referred to as shock and kill (Palmer *et al*, 2011), would involve reactivation of provirus by the use of potent latency-reversing agents (LRAs, the "shock"), and where the infected cells would be subsequently eradicated through cytotoxic T lymphocytes (CTL) or viral-induced cytopathic effects (CP; the "kill"; Deeks, 2012).

Initial shock strategies have included T-cell signaling agonists, such as phorbol esters or related compounds that trigger Ras/NF-κB signaling, or stimulatory cytokines, such as IL-2 and IL-7 (Chun *et al*, 1999; Stellbrink *et al*, 2002; Wang *et al*, 2005). Although these agents are effective in disrupting viral latency, they can cause global T-cell activation, where an extensive amount of cytokine is produced, referred to as a cytokine storm, which makes these agents unfavorable for clinical use. In recent years, most attention has shifted to chromatin-remodeling drugs, such as HDAC and HMT inhibitors, but some remodelers that showed promising results in cell culture produced inconsistent and inconclusive results in clinical trials (Lehrman *et al*, 2005; Sagot-Lerolle *et al*, 2008; Shirakawa *et al*, 2013). Trials using HDAC inhibitors, capable of reactivating HIV-1 provirus *in vitro*, showed that various treatments do not reduce the population size of latently infected cells in patients on antiretroviral therapies. For instance, Vorinostat (SAHA), in single-arm clinical trials where patients received a 400 mg daily dose for 14 consecutive days or 3 days a week for 8 weeks, nor 20 mg Panobinostat, another potent HDACi, tested in a more recent study in which patients were treated three times a week every other week for 8 weeks, did not produce a reduction in the HIV reservoir (Archin *et al*, 2014; Elliott *et al*, 2014; Rasmussen *et al*, 2014). This might be due to inefficiency of the LRA(s) in uniformly inducing the entire population of latent virus, the inability of immune responses in effectively clearing recently activated cells that produce new viral particles, or both (Archin *et al*, 2010, 2014; Routy *et al*, 2012; Rasmussen *et al*, 2014; Spivak *et al*, 2014; Elliott *et al*, 2015; Ke *et al*, 2015; Sogaard *et al*, 2015). Therefore, it is likely that more advanced combination strategies must be used to produce efficient provirus induction for eradication of cells that produce replication-competent viruses using the shock-and-kill strategy.

More recent trials have indicated that combinations of LRAs may produce more robust and global latency reversal *in vivo* (Mbonye & Karn, 2014; Laird *et al*, 2015). In particular, Jiang *et al* (2015) showed the ability of the compound PEP005/ingenol-3-angelate, the active component of the anticancer treatment PICATO (Fidler & Goldberg, 2014), in reversing viral latency can be augmented with both the bromodomain and extraterminal domain (BET) inhibitor JQ1 and P-TEFb agonists (Jiang *et al*, 2015). Additionally, Pache *et al* (2015) have shown that the combined effect of synthetic small molecules (SMAC mimetics) and the HDACi Panobinostat, both used for reactivation of provirus latency in patients, has promising LRA activity. Through targeted RNAi screening, this group identified the SMAC mimetic target as a negative regulator of the non-canonical NF-κB pathway called BIRC2/cIAP1, which also represses HIV-1 transcription. Depletion of BIRC2 by employing SMAC mimetics, which mimic the physiological function of the protein "SMAC/DIABLO", an endogenous BIRC2 antagonist, resulted in reactivation of viral transcription both *in vitro* and *ex vivo* (Pache *et al*, 2015).

In this study, we describe high-throughput screening of synthetic compound libraries for molecules, which can reverse HIV-1 latency, and identified five new compounds that have diverse chemical structures with this activity. The compounds can disrupt latency in cell-based *in vitro* models and importantly from resting CD4$^+$ T lymphocytes isolated from pooled HIV-1-infected patient samples collected from aviremic individuals on antiretroviral therapy. Additionally, we observe significant synergistic effects of our compounds with other LRAs, specifically PEP005. Additionally, *ex vivo* analyses reveal potency of a new combination treatment comprised of one such new compound, referred to as PH02, in combination with PEP005 for reversing HIV-1 latency, demonstrated using a sensitive viral outgrowth assay. Overall, we have identified new LRAs with high potential to reverse HIV-1 latency and have defined an effective new combination treatment for this purpose.

## Results

### HTS of small molecules identifies compounds capable of reversing HIV-1 latency

A primary high-throughput screen (HTS) was performed with ~180,000 small molecules within three separate libraries [Canadian Chemical Biology Network (CCBN), the LIMR Chemical Genomics Center (LCGC), and DIVERSet (ChemBridge)], representing broad chemical diversity. For this purpose, we used the A1 J-Lat Tat-GFP T-cell line with a latent reporter HIV-1 provirus (Fig 1A). In this cell line, expression of the GFP reporter is under control of the HIV-1 LTR promoter such that under basal conditions, in unstimulated cells, GFP expression is not observed (Fig 1B, left panel). However, stimulation by signaling agonists, such as phorbol 12-myristate 13-acetate (PMA), causes induction of GFP expression from the LTR (Fig 1B, right panel), as determined by an Arrayscan Imager.

A pilot screen with 4,761 compounds from a bioactive library "KD2" was initially performed to assess reliability of the assay, using a single-point final concentration of 7 μM for each compound in a 348-well plate format; GFP expression was measured following 24-h treatment. Throughout this study, 50 nM PMA was used as a positive control, and results from the pilot screen were normalized as a percentage activation relative to PMA treatment (Fig EV1A). The calculated Z' factor was higher than 0.5 for the pilot screen, suggesting that the assay was sufficiently robust to be scaled up for a full HTS of larger compound libraries.

The strategy for the HTS process is shown in Fig EV1B. Initially, a final concentration of 7 μM, or 1 μM for the LCGC library, of each compound was tested for activity in the A1 J-Lat cell line, where we observed 104 compounds from the CCBN library that produce > 25% activation, 225 compounds that produced > 40% activation from the LCGC library, and 111 compounds that cause > 40% activation from the DIVERSet, relative to PMA (Fig 1C). By these criteria, the Z' factor for the HTS effort was above 0.5 for all of the libraries. We reconfirmed actives of compounds that produced significant induction, including 38 from the CCBN library, 26 from the LCGC library, and 42 from the DIVERSet (Fig EV1C). Compounds with reconfirmed actives were further tested on

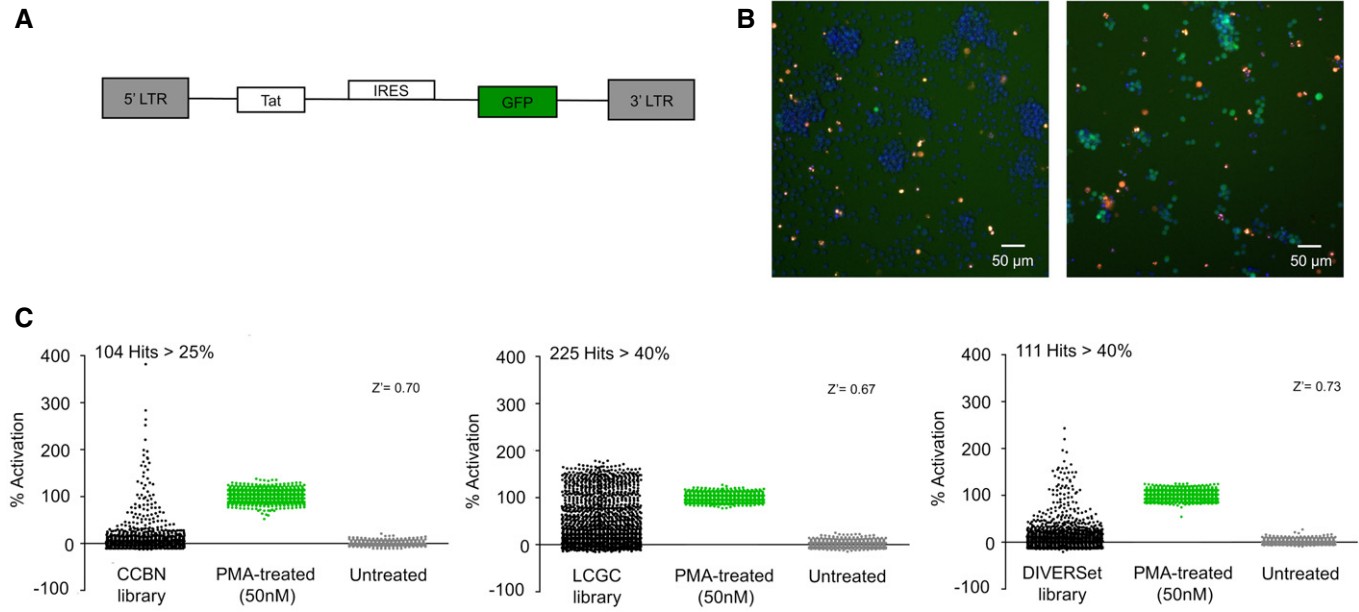

**Figure 1.  High-throughput screening of compounds to identify HIV-1 latency-reversing activity.**

A  Schematic representation of the integrated minivirus (A1 J-Lat Tat-GFP) utilized in the primary screen of small molecules. Expression of the GFP reporter is under control of the HIV-1 LTR promoter in this cell line.

B  Arrayscan images of untreated cells (left panel) or cells treated with PMA for 24 h (right panel; green, GFP; blue, Hoechst; red, PI).

C  Distribution of GFP expression produced by treatment with compounds from the CCBN, LCGC, and DIVERSet libraries (black), or untreated cells (gray). Results are presented as percent GFP expression relative to a positive control reference sample treated with PMA (green) and determined from three biological replicates (mean ± SE, $n = 3$).

parental Jurkat cells lacking the GFP reporter provirus, to eliminate false-positive signals, typically produced by fluorescent properties of the compounds themselves. Compounds that showed activation values for the LTR-GFP reporter, ranging from 30 to 90% relative to PMA in these primary screening assays, were selected for further analysis.

**Five compounds from the primary screens produce concentration-dependent activity**

Activity of compounds identified for latency-reversing activity in the primary screens was further validated by examining the effect of a single concentration (7 μM) on an alternative reporter cell line, bearing an integrated HIV-1 reporter virus where luciferase expression is produced from the 5′ LTR, in cells constitutively expressing the viral TAT protein (Fig 2A), referred to as Jurkat[Tat] LTR-luciferase. Importantly, this cell line has the HIV-1 reporter virus integrated at a different chromosomal location than that of the A1 J-Lat cell line and also enables validation of provirus reactivation with a second enzymatic luciferase assay. In these assays, after 24-h treatment we observed 10 compounds that produced significant induction of luciferase activity, including two compounds from the KD2 library, three from the CCBN library and five compounds from the DIVERSet. Compounds that produced significant effects in these secondary assays were re-ordered from their respective suppliers and examined in more detail using the HIV-luciferase reporter cell line. Among the 10 compounds subjected to further analysis, five, designated PH01, PH02, PH03,

PH04, and PH05, showed considerable luciferase induction after 24-h treatment in a concentration-dependent manner (Fig 2B–F, Table 1, Appendix Fig S1). The maximum responses for the compounds ranged from 28 to 60%, relative to PMA treatment, at effective concentration 50s (EC50) between 0.1 and 5.9 μM (Fig 2B–F). These five compounds represent diverse chemical structures (Fig 2G). In an analysis of time course response, we found that all five compounds produced small effects after only 8-h treatment, and the response continued to increase beyond 48 h (Fig EV2A and B).

**PH compounds do not cause global T-cell activation at concentrations that reactivate latent HIV-1**

A major obstacle to the clinical use of LRAs is that signaling for activation of HIV-1 transcription overlaps that for T-cell activation and that global T-cell activation results in a significant production of inflammatory cytokines, which can produce excessive toxicity (Chun *et al*, 1999; Dybul *et al*, 2002; Wang *et al*, 2005; Bowman *et al*, 2009). Therefore, ideal LRA candidates should be capable of stimulating expression of latent HIV-1 without causing T-cell activation. Two reliable indicators of T-cell activation are the production of IL-2 and expression of CD69 on the T-cell surface. We examined these parameters using a relatively high concentration of each PH compound (7 μM) in the Jurkat[Tat] cell line and found that after 24 h of treatment none of the five compounds significantly increased IL-2 production (Fig 3A) or CD69 expression (Fig 3B).

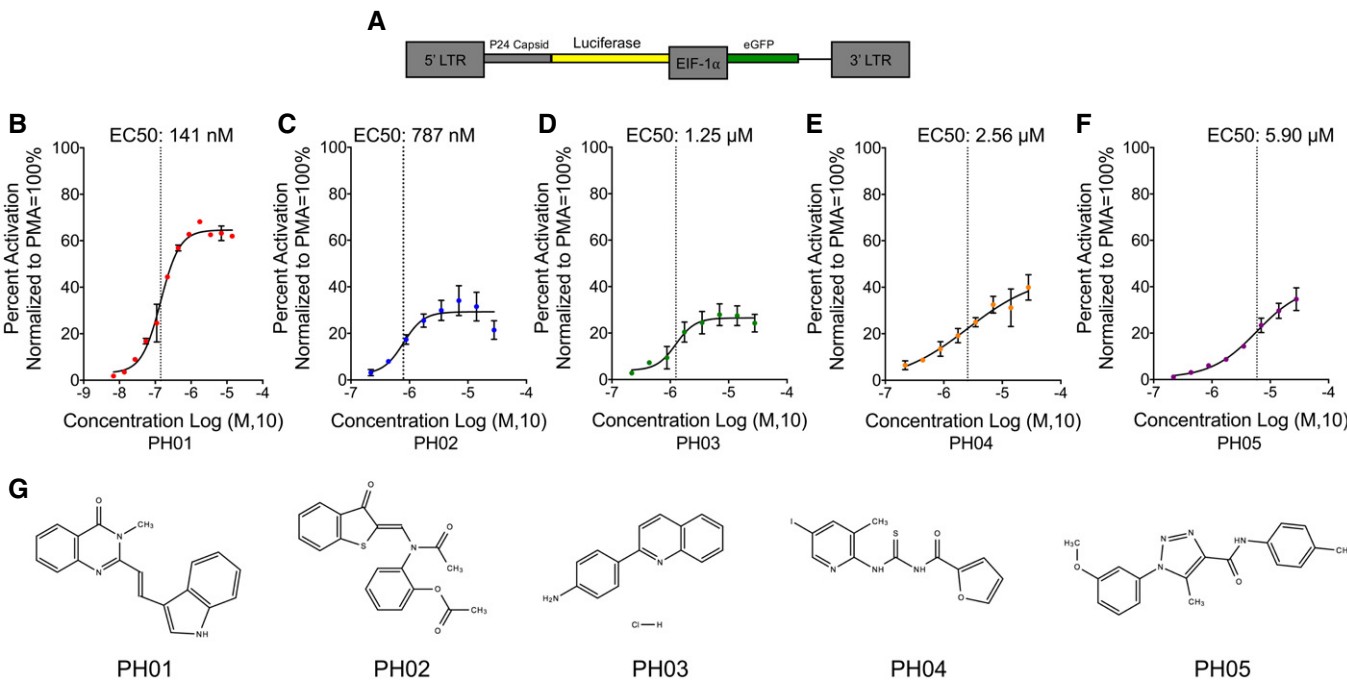

**Figure 2. Concentration–response analysis for five selected PH compounds.**

A    Schematic representation of the pTY-LAI-luciferase reporter virus used in secondary screening assays, where luciferase expression is under the control of the 5′ LTR, in cloned lines derived from Jurkat[TAT] cells bearing single-copy integrants.

B–F    Luciferase activity was measured from reporter cells treated for 24 h with the indicated compound concentrations. Results are presented as percent activation relative to PMA-treated controls. The EC50 for each compound is indicated. Mean and standard error (SE) for the results are determined from three biological replicates (*n* = 3) and technical duplicates.

G    Illustration of chemical structures of the PH compounds.

**Table 1. Details of compounds identified from initial HTS.**

| Name | Library | Vendor | Compound ID | Compound name | M.W | Formula |
|------|---------|--------|-------------|---------------|-----|---------|
| PH01 | DIVERSet | Chembridge | 6000057 | 2-[2-(1H-indol-3-yl)vinyl]-3-methyl-4(3H)-quinazolinone | 301 | $C_{19}H_{15}N_3O$ |
| PH02 | CCBN | Chembridge | 5663181 | 2-{acetyl[(3-oxo-1-benzothien-2(3H)-ylidene)methyl]amino} phenyl acetate | 353 | $C_{19}H_{15}NO_4S$ |
| PH03 | KD2 | Sigma | BF-170 | 2-(4-aminophenyl)quinoline hydrochloride | 256.7 | $C_{15}H_{12}N_2 HCl$ |
| PH04 | DIVERSet | Chembridge | 7727239 | N-{[(5-iodo-3-methyl-2-pyridinyl)amino]carbonothioyl}-2-furamide | 387 | $C_{12}H_{10}IN_3O_2S$ |
| PH05 | DIVERSet | Chembridge | 9103845 | 1-(3-methoxyphenyl)-5-methyl-N-(4-methylphenyl)-1H-1,2,3-triazole-4-carboxamide | 322 | $C_{18}H_{18}N_4O_2$ |

PH01–PH05 produce concentration-dependent responses of HIV-luciferase expression tested on the Jurkat[Tat] LTR-Luciferase reporter cell line. Chemical structures of the compounds are illustrated in Fig 2G.

## PH compounds act synergistically with other LRAs

As described above, the establishment and maintenance of HIV-1 latency involve a combination of factors and cellular regulatory mechanisms. Therefore, a combination of agonists that trigger multiple pathways is likely needed to reactivate the full spectrum of provirus in latently infected cells. To examine this potential for the PH compounds, we examined their activity in combination with four previously characterized LRAs using the Jurkat[Tat] LTR-luciferase cell line. For this purpose, we used LRAs that included the HDAC inhibitor SAHA, the histone methyl transferase inhibitor chaetocin, in addition to ionomycin, and ingenol-3-angelate/ PEP005, which activate NFAT and NF-κB signaling, respectively. These were examined at concentrations previously shown to induce activation of latent HIV-1 in cell lines or primary cells (Bernhard *et al*, 2011; Spina *et al*, 2013; Jiang *et al*, 2015; Laird *et al*, 2015). Among these, both SAHA and PEP005 have FDA approval and are currently being used in various cancer treatments, which makes them ideal candidates for novel combination treatments. As illustrated in Fig 4A–E, all of the PH compounds produced at least a

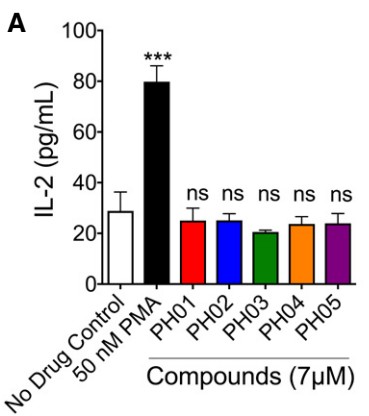
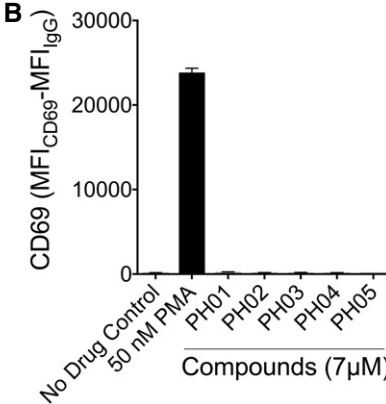

**Figure 3.  Effect of PH compounds on global T-cell activation.**

Jurkat^Tat cells were treated with 7 μM of each compound, 50 nM PMA, or left untreated (no drug control) for 24 h.

A  IL-2 expression was measured in culture supernatants using an ELISA.

B  Cell surface CD69 expression was measured by flow cytometry.

Data information: For statistical analysis, mean and SE for the results are determined from three biological replicates ($n = 3$). Statistical significance, as determined by ratio paired $t$-test, is indicated: ***$P = 0.0008$; ns, no significant difference.

twofold induction in combination with the additional stimuli. Although higher latency-reversal activities were observed for all combinations compared with the single treatments, the highest activity was obtained with 10 nM PEP005 in combination with any of the PH compounds at their EC50, which provided between sixfold and 280-fold induction (Fig 4A–E). To determine whether these effects met the criteria for drug synergy, we applied the Bliss independence model to test the combined effects. Based on this model, a synergistic effect is defined as an observed result that exceeds the predicted additive effect of the compounds administered separately. As defined by these criteria, each of the PH compounds was capable of producing synergy for reactivation of HIV-1 luciferase expression in combination with 10 nM PEP005 (Appendix Fig S2A). Furthermore, at lower concentrations of PEP005, which does not cause reversal of latency on its own (between 1 and 4 nM), combination with the PH compounds caused at least a twofold induction of HIV-luciferase activity (Fig 4F). From Bliss independence analysis, the $\Delta F_{axy}$ calculated for each combination treatment shows values higher than 0, demonstrating that there is significant synergy between the PH compounds in combination with PEP005 (Appendix Fig S2B).

**Compound PH02 reactivates latent HIV-1 integrated at multiple chromosomal locations**

HIV-1 tends to integrate into actively transcribed regions of the host genome (Jordan *et al*, 2003; Lewinski *et al*, 2006; Ikeda *et al*, 2007). Each HIV-1-infected patient carries viral DNA integrated at different chromosomal locations, and as we have previously shown, the site of viral integration influences how latently infected T cells respond to different stimuli (Hashemi *et al*, 2016). Therefore, an ideal LRA should be capable of disrupting latency in cell models carrying integrated viral reporters at multiple different regions within the host genome. To assess this capability for the PH compounds, we used two additional previously described cell lines, designated Jurkat^Tat LTR-DsRed clones #11, and #131, that carry a mini-dual fluorescent HIV-1 LTR

reporter (mdHIV) at different chromosomal locations. The minivirus mdHIV has two fluorescent reporters in which DsRed expression is driven from the 5′ LTR, and eGFP is constitutively expressed from an internal EIF-1α promoter (Fig 5A). With this reporter virus, we typically observe expression of eGFP in unstimulated cells, where viral expression is latent, and where stimulation with PMA causes induction of LTR activity and DsRed expression, producing a shift from the DsRed^−/eGFP^+ population toward double^+ (DsRed^+/eGFP^+) cells, representing active infection (Fig 5B).

Using the mdHIV minivirus reporter virus, we have observed distinct phenotypes produced by the dual reporters in provirus integrated at different chromosomal locations. For example, in cell line Jurkat^Tat LTR-DsRed clone #11, we observe eGFP expression in most unstimulated cells, and ~40% of these produce dsRed expression upon PMA stimulation (Appendix Fig S3A). In contrast, the Jurkat^Tat LTR-DsRed clone #131 has reduced expression of EIF-1α-eGFP in a significant proportion of unstimulated cells, where both reporters appear to have been shut down; however, in these cells, both the LTR and EIF-1α promoters are induced after overnight treatment with 50 nM PMA (Appendix Fig S3B). We selected the clonal cell lines, which produce these distinct expression phenotypes, for further analysis of the latency-reversing activity of the PH compounds. We examined the effect of serial dilutions (28–0.2 μM) of each compound on LTR-DsRed expression after 24-h treatment, and results are presented as Δ mean fluorescence intensity (ΔMFI). Among the five compounds, only PH02 was capable of reactivating viral transcription in both of the cell lines at concentrations above 1.75 μM, and without any apparent cellular toxicity (Fig 5C and D left panels). These data corroborate previous observations that chromosomal integration site strongly influences response to latency-reversing activities and suggest that PH02 might have broad capabilities for reactivation of viral latency. We also tested the effect of PH02 in combination with 10 nM PEP005 in these different clonal cell lines and found concentrations of PH02 higher than 1.75 μM produced increased effects on the HIV-1 LTR transcription with 10 nM PEP005 (Fig 5C and D right panels).

**Figure 4. Effect of PH compounds in combination with additional treatment on HIV-1 LTR expression.**

A–E   The Jurkat[Tat] LTR-luciferase cell line was treated with the EC50/4, EC50, EC50*4 of PH01–PH05, alone or in combination with 300 nM SAHA, 100 nM chaetocin, 1 μM ionomycin, or 10 nM PEP005 as indicated. Luciferase activity was measured after 24 h, and the results are presented as percent activation relative to results from PMA treatment (percent activation for the EC50 of each compound are shown here).

F   The Jurkat[Tat] LTR-luciferase cell line was treated with the indicated concentrations of PEP005 alone (upper panel) and in combination with the PH compounds (EC50; lower panel). Luciferase activity was measured after 24 h, and the results are presented as percent activation described above.

Data information: Mean and SE for the results are determined from three biological replicates ($n = 3$) and technical duplicates. Statistical significance, derived from one-way ANOVA analysis, is indicated as: *$P < 0.05$; **$P < 0.005$; ***$P < 0.0005$; ****$P < 0.00005$. The exact $P$-values are indicated in Appendix Table S1.

## PH02 causes histone modification at the 5′ HIV-1 LTR

To characterize induction of HIV-1 expression caused by PH02 in more detail, we employed chromatin immunoprecipitation (ChIP) assays to examine changes at the LTR promoter using Jurkat[Tat] LTR-DsRed clone #11. We observed that H3K9-3me at the LTR, a chromatin modification typically associated with transcriptional repression, in cells treated with PH02 for 24 h was significantly reduced relative to total histone H3 (Fig 6A left panel), and a similar effect was observed in cells treated with PEP005. Despite that the combination of PH02 and PEP005 produced additive or synergistic effects on expression from the LTR, this combination produced an equivalent reduction in H3K9-3me as either drug treatment separately (Fig 6A left panel). This result suggests that reactivation of HIV-1 expression by these compounds must involve multiple parallel mechanisms in addition to demethylation of repressive chromatin modifications at the promoter.

Accompanying the loss of H3K9-3me, we observe accumulation of H3K9 acetylation on the LTR in cells treated with PH02 and PEP005 for 24 h (Fig 6A right panel). However, the combination of these two compounds caused a significant elevation in H3K9-ac, unlike that observed for loss of H3K9-me3 (Fig 6A). This observation is consistent with the well-defined role of histone acetyl transferase complexes as coactivators of transcription and their known requirement for induction of HIV-1 expression. Despite that PH02 causes accumulation of histone H3K9 acetylation at the LTR, we do not observe a significant effect of this compound on global acetylation or methylation of lysine 9 on H3 in treated cells (Fig 6B, H3K9-ac and H3K9-me3). These observations indicate that the compound likely does not function by inhibiting HDAC or histone methyl transferase activities. Rather, it seems more likely that accumulation of H3K9-ac and loss H3K9 methylation is an indirect consequence of enhanced transcriptional activation by proteins bound to the enhancer region. The HIV-1 LTR is known to be bound by numerous different transcription factors capable of recruiting histone-modifying complexes. Because the PH compounds produce an effect only after 24-h treatment, we reasoned that they could be altering expression of one or more factors, rather than acting as direct signaling agonists. Consequently, we examined expression of NF-κB p65 and SP1 but find their expression to be mostly unaffected by PH02 in the Jurkat[Tat] LTR-DsRed clone 11 line (Fig 6B). We do observe a small increase in expression of SP1 in some cell lines treated with the PH compounds (Fig EV3), but we do not believe this could account for reactivation of provirus because it is not observed in all of the lines where we observe their effect. Consistent with its known effect on PKC–NF-κB signaling, we find that PEP005 causes loss of IκBα after 24-h treatment, alone or in combination with PH02 (Fig 6B). A similar effect is observed in cells treated with the PH compounds, but only after 48 h of treatment (Fig EV3), and consequently, consistent with their synergistic effect with PEP005, we do not believe they could be causing reactivation through upregulation of NF-κB activity on the LTR.

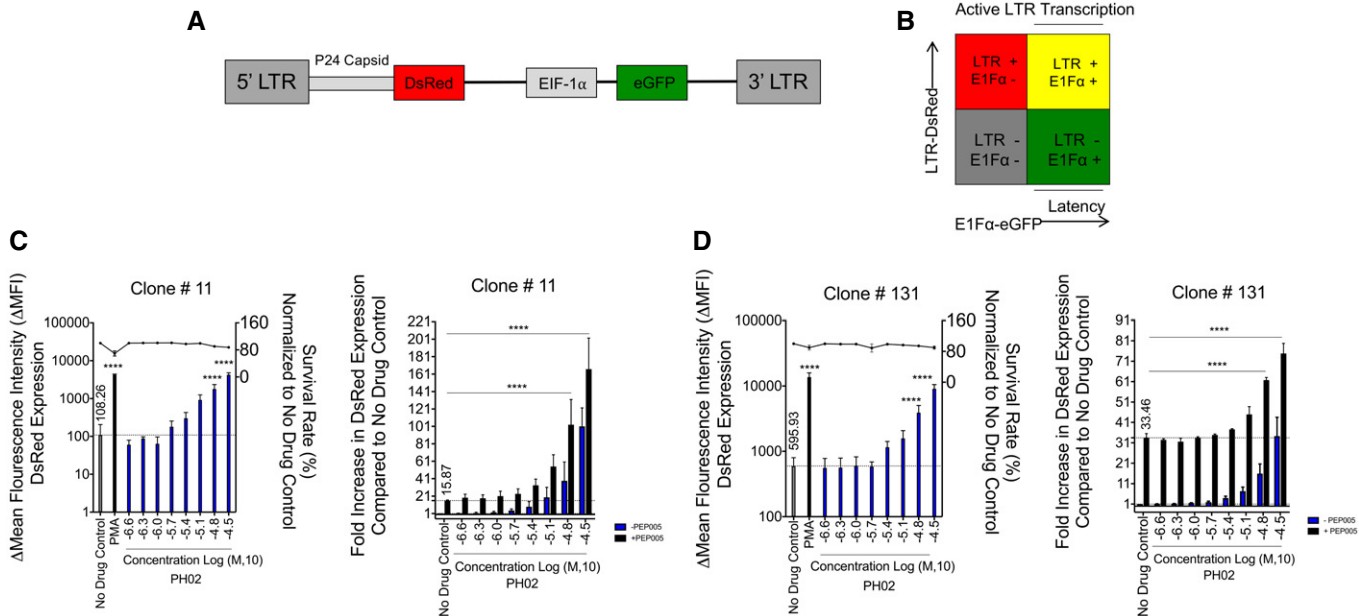

**Figure 5.  Effect of PH02 on HIV provirus integrated at different chromosomal locations.**

A      The effect of PH compounds was analyzed using clonal lines bearing single-copy integrations of the mini-dual fluorescent HIV-1 LTR reporter, md-HIV, where GFP is constitutively expressed from the EIF-1α internal promoter, and DsRed from the 5′ LTR.

B      Fluorescence-activated cell sorting (FACS) of cells bearing md-HIV provirus indicates expression of EIF-1α-GFP and LTR-dsRed. Stimulation of the md-HIV lines generally causes a shift of the fluorescence profile from GFP$^+$/DsRed$^-$ toward GFP$^+$/DsRed$^+$.

C, D   Clone 11 or clone 131 was treated with the indicated concentrations of PH02 (left panels) or PH02 in combination with 10 nM PEP005 (right panels) for 24 h and cells analyzed by flow cytometry. Results presented as Δmean fluorescence intensity (ΔMFI) of DsRed expression and fold increase in DsRed expression for solo and combination treatments, respectively. Mean and SE are determined from three biological replicates ($n = 3$) and technical duplicates. Statistical significance, as determined by one-way ANOVA, is indicated: ****$P < 0.00005$. The exact $P$-values are listed in Appendix Table S1.

## Structural moieties of PH02 required for latency-reversing activity

To probe chemical structure(s) necessary for PH02 to function as a latency-reversing agent, five analogs representing two-dimensional (2D) 75–90% similarities with the parental structure were obtained and tested on the Jurkat$^{Tat}$ LTR-luciferase cell lines and the Jurkat$^{Tat}$ LTR-DsRed clone #11 and #131 cell lines (Fig EV4A, Appendix Table S2). Surprisingly, among these analogs, none showed higher latency-reversal activity than the parental structure with either of the reporter cell lines tested (Fig EV4B and C). Furthermore, PH02$_a$ and PH02$_b$ with the highest 2D similarities (95 and 90%, respectively) to the parent compound showed significant reduction in latency-reversing activity, indicating that omission of the methyl group or both the carbonyl and methyl groups from PH02$_b$ inhibits the ability of PH02 to reverse viral latency. Three additional analogs with 2D similarities of 75–80% did not cause reactivation of viral latency in either of the reporter cell lines (Fig EV4B and C).

## The PH compounds have negligible effects on cell viability

Prior to further analysis of the efficacy for HIV reactivation in primary CD4$^+$ T cells, we measured toxicity of each PH compound on Jurkat$^{Tat}$ cells and peripheral blood mononuclear cells (PBMCs). In both cases, cells were treated with serial dilutions (28–0.1 μM) of

each compound for 24 or 48 h, and examined using an MTT tetra-zolium reduction assay, which measures metabolic activity. The results were normalized to untreated controls and are presented as a survival percentage. Survival rates for both Jurkat$^{Tat}$ cells and primary cells (PBMCs) 24 h post-treatment were above 80% for most compounds at the concentrations used, while reduced viabilities were observed after the 48-h treatment (Fig 7A and B, Appendix Fig S4A). We also examined toxicity produced by the combination of PH compounds with 10 nM PEP005 and other stimuli on PBMCs. No major effects were observed on cell viability after 24 h, except for cells treated with chaetocin in combination with the PH compounds (Fig 7C, Appendix Fig S4B).

## The PH compounds activate viral transcription in resting CD4$^+$ (rCD4$^+$) T cells from HIV-1-infected patients

We next examined the effect of the PH compounds on T cells purified from HIV-1-infected individuals. At the time of enrollment, all participants in the study from which samples were collected had been on antiretroviral therapy a minimum of 6 months, and had viral loads below 50 HIV RNA copies/ml, the clinically accepted limit of detection, and therefore carried only genome-integrated latent provirus.

For this analysis, we purified resting CD4$^+$ (rCD4$^+$) T lympho-cytes from pools of patient samples by two-step negative selection (Appendix Table S3). Purity of the isolated cells was assessed by

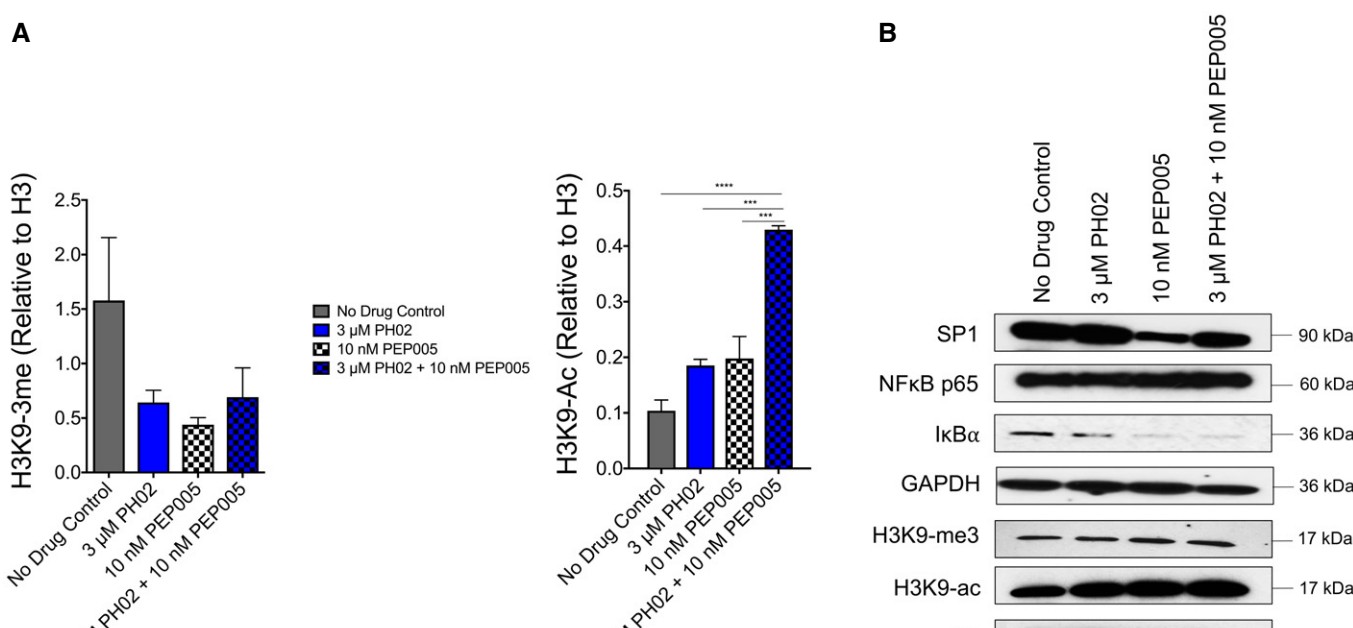

**Figure 6.  Effect of PH02 on histone modification at HIV-1 LTR and global histone modification.**

A   Chromatin immunoprecipitation assays were performed using Jurkat^Tat LTR-DsRed, clone 11. The cells were treated for 24 h with PH02 and/or PEP005, and ChIP was performed using antibodies against histone H3K9-me3 (left panel) and histone H3K9-ac (right panel). The results are presented relative to ChIP with antihistone H3 antibodies. The results are determined from three biological replicates (*n* = 3). Statistical significance, as determined by ratio paired *t*-test, is indicated as follows: ***P < 0.0005; ****P < 0.00005. The exact *P*-values are indicated in Appendix Table S1.

B   Western blot analysis was performed using Jurkat^Tat LTR-DsRed, clone 11, to detect the expression of SP1, NF-κB p65, IκBα, total H3K9-me3, and H3K9ac. The cells were treated for 24 h with PH02 and/or PEP005.

Source data are available online for this figure.

flow cytometry (Appendix Fig S5), which indicated that 96% expressed the CD4 marker but not the activation marker CD69. Purified rCD4$^+$ T cells were treated with 30 μM of each PH compound for 24 h, and RNA from the culture supernatants was analyzed by one-step RT–qPCR, where results are presented as copy numbers of the virion/ml. As shown in Fig 8, all of the PH compounds demonstrated moderate viral latency-reversing activity compared to the PMA-treated positive controls, but among these, PH02 consistently produced the strongest effect. Based on its capability for reactivating latent provirus at multiple chromosomal locations in cell lines, its negligible toxic effect, and that it produces the most robust reactivation of virus from patient samples, we focused our subsequent analysis on this compound (Table 1).

### Identification of minimal concentrations of PH02 and PEP005 that induce T-cell activation

As discussed earlier, an ideal combination of latency-reversing compounds would reverse HIV-1 latency in a broad spectrum of infected cells, without causing global T-cell activation. Accordingly, we examined the effects of PH02 and PEP005 on T-cell activation and toxicity on primary cells (PBMCs) isolated from healthy donors. Previous results have indicated that 10 nM of PEP005 does not cause IL-2 production but does induce upregulation of CD69, in primary cells. Consequently, we examined the effect of lower concentrations of PEP005, alone and in combination with PH02 on

these primary cells. As illustrated in Fig 9, 3 μM of PH02 and 1 nM of PEP005 alone or in combination did not cause significant CD69 induction, whereas PEP005 concentrations higher than 1 nM alone and in combination with PH02 caused expression of CD69 at similar levels as observed for cells treated with PMA (Fig 9). Based on these observations, we expected that 1 nM PEP005 in combination with PH02 should produce negligible toxicity, and have minimal effects on T-cell activation, while being capable of causing significant reversal of HIV latency (Fig 4F, Appendix Fig S2B).

### A combination of PH02 and PEP005 produces HIV latency-reversing activity on CD4$^+$ T cells from HIV-infected patients

Consistent with the above results, we found that 1 nM of PEP005 alone did not cause induction of viral RNA after 24-h treatment of CD4$^+$ T lymphocytes from patients (Fig 10A, Appendix Table S4). However, 1 nM PEP005 in combination with 1 μM PH02 produced a significant response that was double that produced by 1 μM PH02 on its own (Fig 10A). We confirmed this effect using a quantitative viral outgrowth assay (qVOA) with pools of CD4$^+$ T cells isolated from HIV-1-infected patient samples (Fig EV5, Appendix Table S5). Consistent with the results shown above, 1 nM PEP005 alone did not produce reactivation of virus, whereas 1 μM PH02 in combination with 1 nM PEP005 produced significant virus replication (Fig 10B). We conclude that this combination of treatments may represent a promising candidate therapy to target persistent latent viral infection.

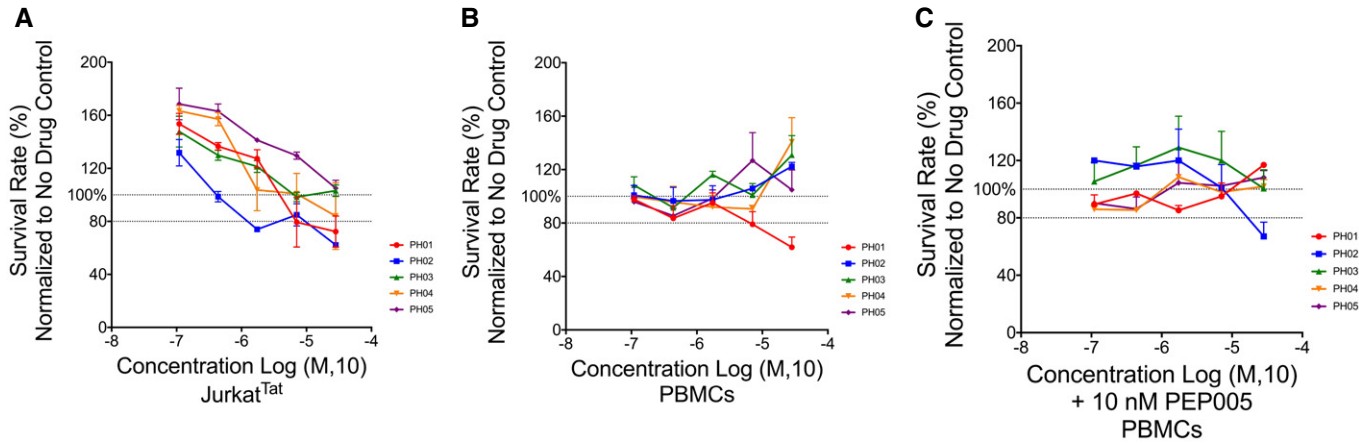

**Figure 7. Effect of PH compounds on cell viability.**

MTT assay was performed to assess cell viability 24 h post-treatment. Results are normalized relative to untreated controls and presented as survival rate percentage. The indicated mean and SE for the results are determined from three biological replicates ($n = 3$) and technical duplicates.

A    Jurkat[Tat] cells were treated with the indicated concentrations of the PH compounds.
B    PBMCs from healthy donors were treated with the indicated concentrations of the PH compounds.
C    PBMCs from healthy donors were treated with the indicated concentrations of the PH compounds in combination with 10 nM PEP005.

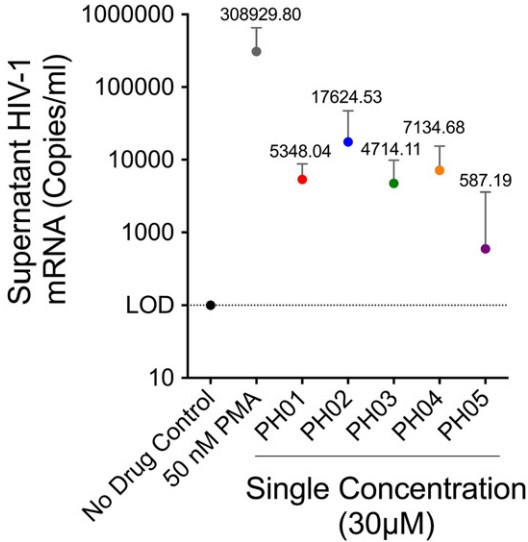

**Figure 8. Effect of PH compounds on expression of HIV provirus in latently infected cells from patients.**

Resting CD4[+] T cells purified from patient samples ($n = 18$, Appendix Table S3) were treated with 30 μM of each PH compound for 24 h, and HIV-1 RNA was measured in culture supernatants. Results are presented as HIV virions/ml (mean ± SE). LOD indicates the limit of detection, 100 copies/ml culture supernatant.

## Discussion

According to recent statistics released by the United Nations Programme on HIV/AIDS (UNAIDS), the number of HIV-infected individuals receiving antiretroviral treatment has increased considerably within the last decade, particularly in African countries, which have the highest number of infected individuals. Effective

antiretroviral treatment can control the virus, allowing patients to enjoy healthier lives and reducing their likelihood to transmit the virus. Despite the many benefits of this treatment, a cure cannot be achieved through HAART and will require a clearer understanding of HIV-1 pathogenesis and the mechanisms contributing to provirus latency. Cells that are latently infected with HIV-1, carrying the stable viral genome, are considered a major obstacle to a cure.

Various technologies have been proposed for eliminating latently infected cells.

In particular, the gene-editing technology CRISPR/Cas9, which is aimed at disrupting genes associated with HIV-1, has recently attracted considerable attention in the field (Ebina *et al*, 2013; Hu *et al*, 2014; Zhu *et al*, 2015; Kaminski *et al*, 2016). To date, this approach has been met with limited success because of the incomplete silencing of genes of interest or interference with the expression profiles of other unintended genes, known as "off-target effects", thus requiring a more in-depth investigation into the safety and feasibility of such applications. Challenges in implementing direct gene manipulation strategies for therapy highlights the fact that the proposed shock-and-kill strategy, which is based on pharmaceutically overcoming mechanisms involved in viral latency, to expose latently infected cells to boosted immune responses or apoptosis, may currently be considered a relatively safer and, more clinically accessible strategy for treating the HIV epidemic. Persistent HIV infection is known to be fueled by the presence of latent HIV-1 reservoirs, but additionally, Lorenzo-Redondo *et al* (2016) have demonstrated active viral replication in sanctuaries of lymphoid tissues in patients on antiretroviral therapies, where the concentration of drugs may not be sufficient to destroy the replicating virus. Thus, clearance of viral reservoirs through the proposed shock-and-kill strategy, along with optimized applications capable of delivering effective drug concentrations more efficiently to sites of active infections, will be necessary for achieving a permanent HIV cure. To this end, the identification of new latency-reversing agents or regulatory factors involved in establishing HIV latency will

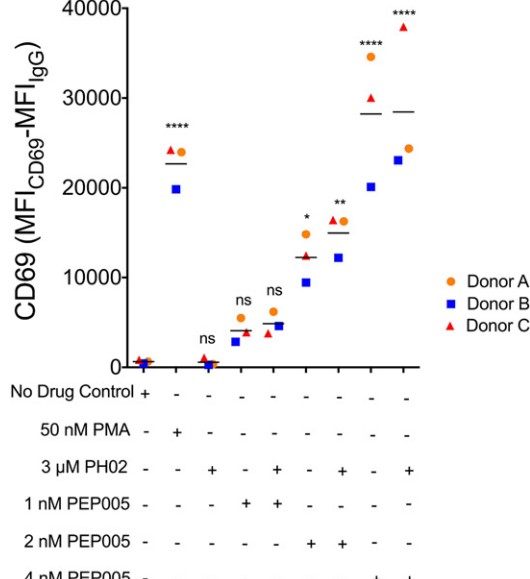

**Figure 9.    Effect of PH02 in combination with PEP005 on T-cell activation.**

Expression of CD69 on PBMCs from three healthy donors (A, B, C; $n = 3$) was measured by flow cytometry following 24-h treatment with 3 μM of PH02 alone or in combination with the indicated concentration of PEP005. Positive control samples were treated with PMA. Statistical significance, as determined by one-way ANOVA, is indicated: $*P < 0.05$; $**P < 0.005$; $***P < 0.0005$; $****P < 0.00005$; ns, no significant difference. The exact $P$-values are indicated in Appendix Table S1.

contribute to ideal therapeutic targets toward the shock-and-kill approach.

Consistent with this aim, this study was initiated to identify new potent latency-reversing agents, with a particular focus on a requirement for global reactivation of latent provirus with synergistic effects in combination with other LRAs. For this purpose, high-throughput screening of small molecules was performed using the A1 J-Lat reporter cell line (Fig 1A). Based on the initial screens, 10 compounds that showed significant latency-reversal activity, compared to the effect of PMA, were selected for follow-up analyses. Five of these, designated PH01, PH02, PH03, PH04, and PH05, reproducibly reactivated viral transcription in a dose-dependent manner when examined using an alternative reporter cell line with an LTR-luciferase reporter (Fig 2B–F). In general, we found that all of the PH compounds from the screen produced synergistic responses for reactivation of HIV in combination with additional latency-reversing compounds, including HDAC and HMT inhibitors and the PKC–NF-κB agonist ingenol-3-angelate/PEP005 (Fig 4A–E). Among these, PEP005 was found to be particularly effective for viral reactivation in combination with our compounds after 24-h treatment (Fig 4, Appendix Fig S2).

Since minimal latency-reversing activity was observed after 8-h treatment with each of the PH compounds in the luciferase reporter cell line and significant induction of viral transcription was only observed at 24 h (Fig EV2A and B), we believe it is unlikely that they act as signaling agonists for pathways downstream of the T-cell receptor (TCR), including those involving protein kinase C (PKC)–NF-κB, MAPK–AP1, or calcineurin–NFAT, which typically produce immediate transcriptional responses. For instance, it has been

previously demonstrated that the latency-reversal activity of PEP005, which was used throughout this study as a second stimulus along with the PH compounds to augment the induction of HIV transcription, is mediated through the PKC–NF-κB signaling pathway, where the phosphorylation of PKCδ/θ and IκBα/IκBε is induced by 12 nM of PEP005 as early as 0.5–2 h post-treatment (Jiang et al, 2015). However, this compound does not cause modulation of NF-κB protein expression, making it clinically more favorable compared to other ingenol esters (Abreu et al, 2014; Jiang et al, 2014). In contrast, the delayed responses demonstrated here by the PH compounds suggest that they may cause upregulation of gene expression or recruitment of one of the many additional transcription factors that bind the HIV-1 LTR but are not the direct targets of T-cell signaling (Pereira et al, 2000). Accordingly, we have examined effects of the compounds on expression of Sp1, which has three binding sites (GC boxes) within HIV-1 LTR, and is known to be required for HIV enhancer activation (Majello et al, 1994), but we do not observe alterations that would be consistent with a role in causing reactivation (Fig 6B). However, it is possible that altered expression of one or more other factors with binding sites on the LTR could result in the delayed transcription activation observed with the PH compounds. Similarly, because of the many additional mechanisms proposed to be involved in HIV latency, including non-coding RNAs and recruitment of repressive complexes by additional sequence-specific binding factors, and DNA methylation, it is also possible that alterations of these more poorly characterized mechanisms might also contribute to delayed responses produced by the compounds.

Several additional observations indicate that the PH compounds are not likely inducing HIV expression by upregulation of NF-κB activity. First, we observe minimal toxicity of the compounds in both the Jurkat[Tat] cell line and PBMCs, even at relatively high concentrations (Fig 7A and B), whereas upregulation of NF-κB protein typically results in excessive cytotoxicity (Baldwin, 2001). Secondly, none of the PH compounds caused IL-2 expression (Fig 3A), which is normally induced in response to T-cell receptor engagement. Similarly, none of the compounds caused CD69 upregulation in either Jurkat[Tat] cells or PBMCs (Figs 3B–9), implying that these agents do not directly trigger NF-κB nuclear translocation, as the CD69 promoter was shown to contain multiple NF-κB binding sites (Lopez-Cabrera et al, 1995).

Other important observations from this study were the effects of both PH02 and PEP005 treatments on chromatin modifications in the HIV-1 promoter region, which were examined in the Jurkat[Tat] LTR-DsRed clone #11 (Fig 6A). Interestingly, cells treated with either PH02 or PEP005 showed a reduction in H3K9 methylation as well as an accumulation of H3K9 acetylation in the HIV-1 LTR relative to the total histone H3 (Fig 6A). PEP005, which is known as a PKC agonist involved in the T-cell signaling pathway, likely causes indirect accumulation of H3K9 acetylation as a consequence of enhanced histone acetyltransferase (HAT) recruitment from transcriptional activation. Consistent with this assumption, it is unlikely that these modifications, accompanied by the PH02 treatment, resulted from a direct inhibition of either HDACs or HMTs by this compound. Indeed, we do not observe alterations in global histone H3K9 acetylation or methylation in cells treated for 24 h (Fig 6B), which supports the possibility that it may promote histone modification at the LTR through mechanisms involving enhanced recruitment of HATs.

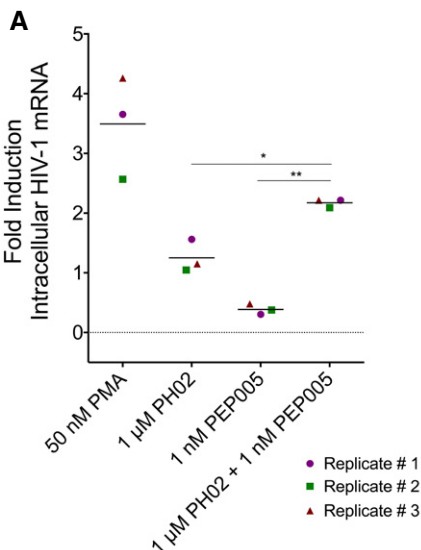

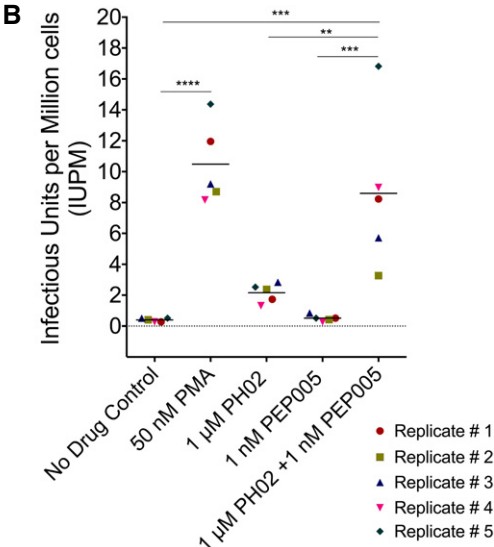

**Figure 10.   Effect of PH02 and PEP005 on expression of HIV provirus in latently infected cells from patients.**

A   Expression of intracellular HIV-1 mRNA was determined from pools of CD4$^+$ T cells from aviremic HIV-infected patients ($n$ = 3, Appendix Table S4) following 24-h treatment as indicated. Results are shown as fold induction relative to untreated controls.

B   Quantitative viral outgrowth assays (qVOA) were used to measure reactivation of HIV-1 provirus from latency in pools of cells from patients. Cells were treated with the indicated concentrations of PH02 and/or PEP005 for 24 h, prior to analysis of virus production. As indicated in Appendix Table S5, five different CD4$^+$ T-cell pools ($n$ = 5) were used for this analysis.

Data information: Statistical significance was determined by one-way ANOVA and indicated as: *$P$ < 0.05; **$P$ < 0.005; ***$P$ < 0.0005; ****$P$ < 0.00005. The exact $P$-values are listed in Appendix Table S1.

We have previously shown, using *in vitro* analyses of clonal cell lines carrying the mini-dual HIV reporter virus at different chromosomal locations, that the epigenetic landscape at the site of integration has a significant effect on the efficacy of various LRAs in disrupting latency. Consistent with these findings, we observed different responses to the PH compounds using four latently infected reporter cell lines bearing the virus at different genomic locations. Importantly, however, we demonstrated that PH02 is capable of reactivating viral transcription in all of the reporter cell lines we examined. These observations support the contention that a combinational regimen targeting multiple distinct pathways involved in the establishment of viral latency would be required for global stimulation of latent proviruses integrated at various sites within the genome, as different provirus species exhibit a unique sensitivity to signaling agonists and chromatin-remodeling agents.

The highest latency-reversing activity associated with PH02, presumably independent of the proviral integration site, which was observed alone and in combination with PEP005 through our intensive *in vitro* analyses, led us to further investigate the effect of these two compounds on CD4$^+$ T lymphocytes purified from HIV-1-infected individuals on HAART. The ability of PH02 to significantly resume HIV-1 transcription at a clinically relevant concentration in combination with only 1 nM of PEP005 was initially demonstrated by measuring HIV-1 intracellular RNA 24 h post-treatment (Fig 10A). This finding was further confirmed by performing a viral outgrowth assay quantifying only the replication-competent provirus (Fig 10B).

In conclusion, in this report we describe a potentially novel combination of treatments that may be used to uniformly induce HIV transcription in a broad range of latently infected cells. We

suggest that this combination may be a candidate for application with the shock-and-kill strategy to expose reactivated cells for the purging of latent infection.

# Materials and Methods

### High-throughput screening of compounds

In total, 180,000 small molecules represented by the KD-2, CCBN, DIVERSet, and LCGC libraries (CDRD) were screened in 384-well plate format. For the screen, 12,500 reporter cells were resuspended in phenol red-free RPMI with 10% heat-inactivated FBS and dispensed into wells. Each compound was pinned in the wells using the PlateMate employing FP3 pins for a final concentration between 1 and 7 μM, depending on library, and the plates were then incubated overnight at 37°C, 5% $CO_2$. 10 μl of 1 μg/ml Hoechst and 500 ng/ml propidium iodide (Hoechst 33342 Trihydrochloride, Invitrogen # H3570 and PI, Invitrogen # P1384MP) were added to wells and incubated at 37°C, 5% $CO_2$ for 30 min. The plates were then scanned using a Cellomics Arrayscanner (Thermo Scientific) with three channels to measure Hoechst, GFP, and PI. The results are presented as a percent activation of the GFP reporter normalized to the positive control (50 nM PMA), such that % activation = [(X − C$^-$)/(C$^+$ − C$^-$)] × 100; X = individual well reading where C$^-$ = the average negative control reading, and C$^+$ = the average positive control reading.

Compounds, including structural analogs of the compound PH02, were purchased from Chembridge (San Diego, CA, USA),

Maybridge (Cambridge, UK), LIMR Chemical Genomics Center (Wynnewood, PA, USA), and Sigma-Aldrich. The concentrations EC50/4, EC50, and EC50*4 stated throughout this study have been calculated based on the EC50 value obtained from dose–response analysis for each PH compound using the Jurkat[Tat] LTR-Luciferase cell line. Unless otherwise indicated, suberoy-lanilide hydroxamic acid (SAHA; Toronto Research Chemical) was used at 300 nM, chaetocin (Sigma) at 100 nM, ionomycin (Sigma) at 1 μM, and ingenol-3-angelate/PEP005 (Cayman Chemical) at 1–10 nM.

**Recombinant DNA molecules, cell lines, and cell culture**

Clonal cell lines bearing the pTY-LAI-luciferase and mini-dual HIV-1 reporter virus were described previously (Bernhard *et al*, 2011, 2013). J-Lat Tat-GFP (A1) cells, carrying an integrated HIV-1 reporter LTR-Tat-IRES-GFP, were obtained from the NIH-AIDS reagent resource. Cell lines and healthy PBMCs were cultured in RPMI 1640 (Sigma-Aldrich) supplemented with 10% FBS, 100 U/ml penicillin, 100 μg/ml streptomycin, and 0.8 mg/ml Genetecin (G418 sulfate) and incubated in a humidified 37°C and 5% $CO_2$ atmosphere. Primary $CD4^+$ T lymphocytes were cultured in Super T-cell media (STCM) consisting of RPMI 1640 supplemented with 10% heat-inactivated FBS, 100 U/ml IL-2, 100 U/ml penicillin, 100 μg/ml streptomycin, and 2% conditioned culture supernatant (T-cell growth factor, TCGF) from healthy PBMCs treated with 2 μg/ml phytohemagglutinin (PHA) and 5 ng/ml PMA for 4 h.

**Reporter assays, ELISA, and flow cytometry**

Luciferase assays were performed in 96-well plate format, where 100 μl of cell culture containing $1 \times 10^5$ luciferase reporter cells was plated per well. After the indicated time of treatment, 100 μl of luciferase assay reagent (Superlight™ luciferase reporter Gene Assay Kit; BioAssay Systems) was added per well, and incubated for 2 min at RT, when luciferase activity was measured on a Victor™ X3 Multilabel Plate Reader. Results were presented relative to the positive control samples treated with 50 nM PMA. T-cell activation was measured using $1 \times 10^5$ wild-type Jurkat[Tat] cells or PBMCs purified from healthy donors, treated as indicated. After 24 h, cells were incubated with 2.4G2 antibody to block nonspecific binding and further stained with 5 μl of anti-human CD69 or an IgG1 isotype control antibody conjugated with PE-Cy™7. After 30-min incubation on ice, the cells were washed twice with PBS and analyzed using a Becton Dickinson LSRII flow cytometer (BD Biosciences) to determine mean fluorescence intensity (MFI), using FlowJo software. The mean fluorescence intensity (MFI) obtained from cells stained with IgG isotype antibody control was subtracted from that from cells stained with CD69 to compensate for background ($MFI_{CD69}$-$MFI_{IgG}$). IL-2 production was measured in cell culture supernatants using a human IL-2 ELISA System (eBioscience). Jurkat[Tat] clones bearing integrations of the mini-dual HIV reporter virus, clone #11 and #131, were plated at $1 \times 10^5$ per well in a 96-well plate format, treated for the indicated time, washed with 1 ml PBS, and then transferred into flow cytometry tubes, where they were fixed with freshly made 1% formaldehyde for 10 min at RT prior to analysis by flow cytometry. Results are presented as ΔMFI, where MFI obtained from parental

Jurkat[Tat] cells undergoing the same treated was subtracted from the MFI corresponding to each cell line; ΔMFI = (MFI compound-treated cell line 131/11 − MFI compound-treated WT Jurkat[Tat] cells).

We applied the Bliss independence model to assess the latency-reversing activity of drug combinations, which was calculated by the equation $f_{axy,P} = f_{ax} + f_{ay} - (f_{ax})\ (f_{ay})$, where $f_{ax}$ = fraction affected, drug $x$ and $f_{ay}$ = fraction affected, drug $y$. The experimentally observed fraction affected ($f_{axy,O}$) was compared with the calculated predicted value ($f_{axy,P}$) and $\Delta f_{axy} = f_{axy,O} - f_{axy,P}$ if the compounds produced an additive effect. Based on this model, an $\Delta f_{axy}$ greater than 0 ($\Delta f_{axy} > 0$) indicates a synergistic effect when administrated together.

**Chromatin immunoprecipitation (ChIP) assays**

Jurkat[Tat] LTR-DsRed cells, clone #11, were treated with 3 μM PH02 and/or 10 nM PEP005, for 24 h, and the cells cross-linked with 1% formaldehyde for 10 min at RT. The cells were resuspended in cold NP-40 lysis buffer (0.5% NP-40, 10 mM Tris–HCl pH 7.8, 3 mM $MgCl_2$) containing freshly added protease inhibitor cocktail (PIC, Sigma-Aldrich) and spun to collect nuclei. The nuclei were sonicated in buffer containing 10 mM Tris–HCl (pH 7.8), 10 mM EDTA, and 0.5% SDS to obtain DNA fragments of 200–2,000 bp using a Bioruptor sonicator (Diagenode). The chromatin fractions were precleared with protein A/G agarose beads (Millipore) and immune-precipitated with anti-H3K9-me3, anti-H3K9-ac, or anti-H3 antibodies (Abcam) according to the manufacturer's instructions. Cross-links were reversed by incubating the samples at 68°C for 2 h, and DNA was purified using QIAquick PCR columns and assayed by qPCR using primers specific to the HIV-1 LTR core promoter, AGTGGCGAGCCCTCAGAT, and AGAGCTCCCAGGCTCAAATC. Results are presented as normalized relative to that for histone H3 immuno-precipitates.

**Immunoblot analyses**

$5 \times 10^6$ luciferase reporter cells/Jurkat[Tat] LTR-DsRed, clone 11, were treated with the indicated concentration of the compounds. After treatment, the cells were lysed in RIPA buffer (50 mM Tris–HCl, pH 8.0, 150 mM NaCl, 1% NP-40, 0.5% sodium deoxycholate, 0.1% SDS), supplemented with 1× protease inhibitor cocktail and 1 mM DTT, to collect the whole-cell protein extract, which was further quantified using Bradford assay. A 30 μg equivalent of the protein extract was mixed with the 5× SDS–PAGE sample buffer and boiled for 5 min, and then, it was separated using 12% SDS–PAGE gel and transferred to a nitrocellulose membrane. The membrane was blocked with 3% milk (w/v) in TBS for 1 h and then incubated with a primary antibody overnight at 4°C. The antibodies used were as follows: SP1—Abcam 13370 (1:4,000), NF-κB, p65—Abcam 7970 (1:1,000), IκBα—Abcam 32518 (1:5,000), GAPDH—Abcam 9484 (1:4,000), H3K9-acetylation—Abcam 10812 (1:500), H3K9-trimethy-lation—Abcam 8898 (1:500), and H3—Abcam 1791 (1:5,000). The membrane was washed three times with TBS and then incubated with the HRP-conjugated goat secondary antibody at RT. After incubation, the membrane was washed with TBS, and the signal was developed with the SuperSignal West Femto chemiluminescent substrate (Thermo Fisher).

## MTT cell viability assays

$1 \times 10^5$ Jurkat$^{Tat}$ cells or PBMCs were plated in 96-well plates and treated as indicated. The plates were incubated at 37°C, 5% $CO_2$ for 24 or 48 h, and then, 20 μl of 5 mg/ml MTT reagent [3-(4,5-dimethylthiazol-2-yl)-2,5 diphenyltetrazolium bromide, Sigma-Aldrich] was added per well and the plates incubated a further 4 h at 37°C. 100 μl of 10% SDS and 0.01 M HCl was then added, the plates incubated overnight at 37°C, and the A595 was determined for each well using a Victor™ X3 Multilabel Plate Reader (Perki-nElmer). Results were normalized to an untreated control and presented as survival rate %.

## Primary cells and patient samples

Peripheral blood mononuclear cells were isolated from healthy donors by density centrifugation on a Ficoll-Hypaque gradient (Ficoll-Pague™ Plus, GE Healthcare). HIV-infected patient samples were provided through the BC Centre for Excellence in HIV/AIDS at St. Paul's Hospital, Vancouver. All participants had been on antiretroviral therapy for a minimum of 6 months with plasma HIV-1 RNA levels < 50 copies/ml. All subjects provided written, informed consent prior to their inclusion in the study and the experiments conformed to the principles set out in the WMA Declaration of Helsinki and the Department of Health and Human Services Belmont Report.

CD4$^+$ T lymphocytes were isolated using the EasySep™ Human CD4 Positive Selection Kit (Stem Cell, Cat # 18052), and further purified by negative depletion of cells expressing CD69, CD25, or HLA-DR using Miltenyi magnetic beads (Miltenyi Biotec). The purity of isolated T cells was assessed by staining with FITC-conjugated monoclonal antibody against CD4 (BD Pharmingen™) and PE-Cy-conjugated monoclonal antibodies against CD69 (BD Pharmingen™) and analysis by flow cytometry.

## Measurement of HIV-1 replication from patient samples

HIV-1 RNA was measured from $1 \times 10^6$ resting CD4$^+$ T lymphocytes treated with 30 μM of each compound. Extracellular RNA was isolated using the ZR Viral RNA Kit (Zymo Research) and total cellular mRNA using the RNeasy MinElute cleanup kit (Qiagen). HIV-1 RNA concentrations were assayed in one-step RT–qPCRs using the iTaq™ universal SYBR Green one-step kit (BIO-RAD) with the HIV-1 LTR-specific primers, AGCCGCCTAGCATTTCATC and CAGCGGA AAGTCCCTTGTAG. RNA copy number was calculated using a standard curve generated from titrated infections with RGH-PGK virus (Dahabieh *et al*, 2013). Reproducibility of RT–qPCR analysis was confirmed using a second set of primers within HIV-1 p24 Gag, AGCAGCCATGCAAATGTTA, AGAGAACCAAGGGGAAGTGA. HIV-1 RNA determinations were normalized to GAPDH RNA measured using the primers CAGCCTCAAGATCATCAGCA and TGTGGTCATG AGTCCTTCCA.

For quantitative viral outgrowth assays (qVOA), serial dilutions of CD4$^+$ T lymphocytes from pooled HIV-infected patients were plated in 6- or 24-well plates ($1 \times 10^6$ cells or $< 1 \times 10^6$ cells, respectively). The cells were treated as indicated for 24 h, when the media were removed and replaced with fresh media. Following an additional incubation for 5 h, $4 \times 10^6$ or $1 \times 10^6$ MOLT4/CCR5 cells

**The paper explained**

**Problem**

Despite over 30 years of intensive research in the HIV/AIDS field, there is still no cure. The major challenge in developing a cure is associated with the population of cells carrying a stable and silent integrated virus within its genome (provirus), representing a latent reservoir. This population is unrecognizable to both the patient's immune system and current antiretroviral therapies, and upon cessation of therapy, viral particles reappear in the blood stream, causing disease progression and eventually AIDS development.

**Results**

This study describes the discovery of five novel compounds designated PH01–PH05, which have the ability to reawaken proviruses from the latent reservoir without causing cytotoxic effects. Through in-depth investigation of these compounds, the research identifies a new combinational regimen comprised of PH02 and a previously known compound, PEP005, which exhibits significant latency-reversing activity when examined in both *in vitro* cell models and CD4$^+$ T cells purified from HIV-infected patients on antiretroviral therapy.

**Impact**

An HIV cure could be achieved through elimination of the latent reservoir of infected cells. One possibility toward this goal known as shock and kill involves forcing reactivation of the latent reservoir by therapeutic intervention to expose those infected cells to immune responses or virus-induced apoptosis. We believe the combinatorial regime identified in this study may provide a novel means to abolish the HIV-persistent infection in a patient's body.

were added to the 6- or 24-well plates, respectively. The plates were incubated for a further 14 days and the media changed every 5 days, when the supernatants were collected and filtered through a 0.45-μm filter unit to remove cell debris. Viral RNA was isolated from the culture supernatants and analyzed as described above. The number of latently infected cells was determined from the initial number of CD4$^+$ T cells in the assay, the number of replicates, and the number of positive outcomes, using the IUPMStats V1.0 infection frequency calculator (http://silicianolab.johnshopkins.edu) to produce a "maximum-likelihood estimation (MLE)" (Laird *et al*, 2013).

## Statistics

For statistical analysis, mean and standard error (SE) for the results were determined from three biological replicates and technical duplicates, unless otherwise stated. Statistical significance analysis involved the ratio paired *t*-test or one-way ANOVA, as indicated in the figure legends.

**Expanded View** for this article is available online.

## Acknowledgements

This research was supported by grants from the Canadian Institutes for Health Research (F16-01210, HOP-134066, and HOP-143172) to I.S. We thank Jack Yang and Tracy Luong for assistance with the high-throughput screen, and Mel Reichman and Scott Donover (Lankenau Institute for Medical Research) for the screening data analysis and provision of compounds.

## Author contributions

PH conceived and designed the *in vitro* and *ex vivo* experiments, performed the experiments, analyzed the data, and wrote and revised the manuscript. KB, WB, and NH designed and performed the high-throughput screening. AL edited the manuscript. TAP supervised the HTS and edited the manuscript. PRH coordinated the HIV-infected patient samples and edited the manuscript. IS supervised the experiments, and wrote and revised the manuscript.

## Conflict of interest

The authors declare that they have no conflict of interest.

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
