## [Review Process File · EMBO Molecular Medicine]

Compounds Producing an Effective Combinatorial Regimen For Disruption of HIV-1 Latency

Pargol Hashemi, Kris Barreto, Wendy Bernhard, Adam Lomness, Nicolette Honson, Tom A. Pfeifer, P. Richard Harrigan and Ivan Sadowski

Review timeline:

Submission date:	22 June 2017
Editorial Decision:	19 October 2017
Revision received:	10 November 2017
Editorial Decision:	17 November 2017
Revision received:	20 November 2017
Accepted:	20 November 2017

Editor: Céline Carret

Transaction Report:

1st Editorial Decision

19 October 2017

Thank you for the submission of your manuscript to EMBO Molecular Medicine. We have now heard back from the three referees whom we asked to evaluate your manuscript.

As you will see from the comments below, the three referees are enthusiastic about the study, even though recommendations are made to provide more mechanistic insights into the mode of action, that I would like to encourage you to address, at least in part.

We would welcome the submission of a revised version within three months for further consideration. Please note that EMBO Molecular Medicine strongly supports a single round of revision and that, as acceptance or rejection of the manuscript will depend on another round of review, your responses should be as complete as possible.

I look forward to receiving your revised manuscript.

***** Reviewer's comments *****

Referee #1 (Comments on Novelty/Model System):

The work is interesting and correctly performed. The authors first screened about 180,000 compounds to identify molecules reactivating a latent HIV provirus in a model cell line. They selected 5 compounds that were further characterized, either alone or in combination with known molecules. One compound, termed PH02, reactivates viral replication in CD4+ T cells from HIV-1 infected individuals under successful antiretroviral treatment. The compound works synergistically with previously characterized compounds, including PEP005. The novelty resides in the identification of PH02 and other compounds, and their synergistic activity against the viral reservoir.

Referee #2 (Comments on Novelty/Model System):

The manuscript by Hashemi et al. deals with the issue of identifying compounds that, in combination, could lead latent HIV-1 proviruses out of their latent state. In particular, the authors have identified 5 compounds by high throughput screening (HTS) that reverse latent HIV-1 infection without causing a generalized T cell activation. These novel compounds synergize with previously identified latency reversing agents (LRA), particularly with Ingenol in both cell lines and primary resting CD4+ T cells isolated from pts. receiving cART showing synergy by the standard Q-VOA in the case of the combination between PH02 and PEP005.

The paper is very clearly written and represents an impressive thorough analysis of candidate novel LRA up to their validation in resting CD4 T cells of patients under suppressive cART.

Referee #2 (Remarks):

impressive work! it would be impossible to reduce it to a short report. The main limitation is the concept of "shock and kill" as a whole; however, within this hypothetical model this study has been really well conducted.

Referee #3 (Comments on Novelty/Model System):

Phenotypic screen for novel latency reactivating agents is well performed and well described.

Referee #3 (Remarks):

Authors report on a phenotypic screen for novel latency reactivating agents. The screen is performed well and the hits are interesting. The paper is very well written. Unfortunately, although some epigenetic characteristics are tested, the true mechanism of action of the novel compounds is not elucidated. I find this a prerequisite for a paper in EMBO Molecular Medicine.

1st Revision - authors' response

10 November 2017

Referee #1: The work is interesting and correctly performed. The authors first screened about 180,000 compounds to identify molecules reactivating a latent HIV provirus in a model cell line. They selected 5 compounds that were further characterized, either alone or in combination with known molecules. One compound, termed PH02, reactivates viral replication in CD4+ T cells from HIV-1 infected individuals under successful antiretroviral treatment. The compound works synergistically with previously characterized compounds, including PEP005. The novelty resides in the identification of PH02 and other compounds, and their synergistic activity against the viral reservoir.

Referee #2: The manuscript by Hashemi et al. deals with the issue of identifying compounds that, in combination, could lead latent HIV-1 proviruses out of their latent state. In particular, the authors have identified 5 compounds by high throughput screening (HTS) that reverse latent HIV-1 infection without causing a generalized T cell activation. These novel compounds

synergize with previously identified latency reversing agents (LRA), particularly with Ingenol in both cell lines and primary resting CD4+ T cells is olated from pts. receiving cART showing synergy by the standard Q-VOA in the case of the combination between PH02 and PEP005.

The paper is very clearly written and represents and impressive thorough analysis of candidate novel LRA up to their validation in resting CD4 T cells of patients under suppressive cART.

(Remarks): impressive work! it would be impossible to reduce it to a short report. The main limitation is the concept of "shock and kill" as a whole; however, within this hypothetical model this study has been really well conducted.

Referee #3: Phenotypic screen for novel latency reactivating agents is well performed and well described.

(Remarks): Authors report on a phenotypic screen for novel latency reactivating agents. The screen is performed well and the hits are interesting. The paper is very well written. Unfortunately, although some epigenetic characteristics are tested, the true mechanism of action of the novel compounds is not elucidated. I find this a prerequisite for a paper in Embo Molecular Medicine.

We thank all three referees for their encouraging comments; several of the referees commented that this was "impressive work" and overall the screen was well performed. All of the referees have commented that the writing was good, and therefore we have made only minor corrections to spelling and grammar in most of the text throughout the revisions. Referee #2 alludes to "limitations of shock and kill", but without mentioning specifics in their comments. We, like others working in this field, recognize the steep incline that potential shock and kill therapies face for effective treatment, and consequently we have noted this in the Introduction in the paragraph ending "Therefore, it is likely that more advanced combination strategies must be used to produce efficient provirus induction for eradication of cells that produce replication-competent viruses using the shock and kill strategy."

Referee #3, notes that we have not described a mechanism for reactivation of HIV latency by the compounds. We believe it important to point out that this study represents the first analysis of completely synthetic compound libraries where novel compounds were identified that can reactivate HIV latency in cell lines, as well as infected PBMCs from patients, and importantly can also cause induction of replication competent virus from latently infected samples from patients. One previous study describes a screen using a similar number of compounds (~200,000), but this study failed to identify compounds capable of reactivating expression of replication competent virus from patient samples (Micheva-Viteva et al., 2011). Furthermore, we note that most comparable previous screens were performed with much smaller "known drug" libraries or libraries of natural products where mechanistic activity had already been identified or at least suspected.

We agree that it will be important to understand the biochemical effect(s) of the compounds we have identified, and we have spent considerable effort over the past several months towards this goal. Unfortunately none of the most obvious possible mechanisms that could reactivate HIV from latency seem to be affected. We provide additional evidence in the revised manuscript showing that PH02 does not affect global histone acetylation and also does not affect expression of key transcription factors regulating activation from the HIV enhancer. In additional experiments, not described in the manuscript, we have determined, unfortunately, that none of the PH compounds inhibit growth or other obvious phenotypes in yeast. From these efforts we believe that the compounds must affect previously uncharacterized mechanisms for maintenance of viral latency. Identification of mechanism of action for compounds identified in synthetic small molecule libraries is never trivial, and will require detailed analysis of responses using proteomics and genomics, which seems beyond the scope of the present manuscript.

2nd Editorial Decision

17 November 2017

Thank you for the submission of your revised manuscript to EMBO Molecular Medicine. We have now received the enclosed reports from the referees that were asked to re-assess it. As you will see

the reviewers are now globally supportive and I am pleased to inform you that we will be able to accept your manuscript pending final amendments.

***** Reviewer's comments *****

Referee #1 (Remarks for Author):

The manuscript has been improved and the authors have addressed my concerns

Referee #3 (Remarks for Author):

The lack of MOA dampens my enthusiasm but the work deserves publication so the work can be shared and continued.

Corresponding Author Name: Dr. Ivan Sadowski
Journal Submitted to: EMBO Molecular Medicine
Manuscript Number: EMM-2017-08193